# Innovative Dynamic Ultrasound Diagnosis of First Rib Stress Fracture in an Adolescent Athlete—A Case Report

**DOI:** 10.3390/diagnostics15192437

**Published:** 2025-09-24

**Authors:** Yonghyun Yoon, King Hei Stanley Lam, Chanwool Park, Jaeyoung Lee, Jangkeun Kye, Hyeeun Kim, Seonghwan Kim, Junhan Kang, Anwar Suhaimi, Teinny Suryadi, Daniel Chiung-Jui Su, Kenneth Dean Reeves, Stephen Cavallino

**Affiliations:** 1Department of Orthopaedic Surgery, Gangnam Sacred Heart Hospital, Hallym University College of Medicine, 1 Singil-ro, Yeongdeungpo-gu, Seoul 07441, Republic of Korea; 2Incheon Terminal Orthopedic Surgery Clinic, Inha-ro 489beon-gil, Namdong-gu, Incheon 21574, Republic of Korea; humanpcw94@gmail.com (C.P.); 2wo02wo0@naver.com (J.L.); 3International Academy of Regenerative Medicine, Namdong-gu, Incheon 21574, Republic of Korea; 4Board of Clinical Research, International Association of Musculoskeletal Medicine, Kowloon, Hong Kong; anwar@ummc.edu.my (A.S.); painfreedoc22@gmail.com (T.S.); 5The Faculty of Medicine, The University of Hong Kong, Pokfulam, Hong Kong; 6The Faculty of Medicine, The Chinese University of Hong Kong, New Territories, Hong Kong; 7Rolfing Spine Posture Institute, Seoul 06236, Republic of Korea; chirojk@naver.com; 8Korean Association of Cyriax Orthopaedic Medicine, 375-1 Yongmun-ro, Yongmun-myeon, Yangpyeong 12522, Republic of Korea; pilateshe@naver.com (H.K.); yeskimmy@daum.net (S.K.); chirodrkang@naver.com (J.K.); 9Department of Rehabilitation Medicine, University Malaya, Kuala Lumpur 50603, Malaysia; 10Department of Physical Medicine and Rehabilitation, Hermina Podomoro Hospital, North Jakarta 14350, Indonesia; 11Department of Physical Medicine and Rehabilitation, Medistra Hospital, South Jakarta 12950, Indonesia; 12Physical Medicine and Rehabilitation, Synergy Clinic, West Jakarta 11510, Indonesia; 13Department of Physical Medicine and Rehabilitation, Chi Mei Medical Center, Tainan 710, Taiwan; dr.daniel@gmail.com; 14Tempo Regeneration Center for Musicians, Tainan 700, Taiwan; 15Independent Researcher, Roeland Park, KS 66205, USA; deanreevesmd@gmail.com; 16European School of Prolotherapy (ESP), 1st Mednikarska Str., 1510 Sofia, Bulgaria; s.cavallino@gmail.com; 17Hackett Hemwall Patterson Foundation (HHPF), 7880 Sweeny Rd., Barneveld, WI 53507, USA

**Keywords:** ultrasound diagnosis, first rib stress fracture, adolescent athletes, overhead throwing injury, scapular winging, dynamic ultrasound, enthesopathy

## Abstract

**Background:** First rib stress fractures (FRSFs) are exceptionally rare in skeletally immature athletes and are frequently overlooked because their symptoms mimic more common scapular conditions such as scapular dyskinesis or thoracic outlet syndrome. Early and accurate identification is critical to avoid delayed union, prolonged disability, and misdirected management. **Case Presentation:** We report a 12-year-old elite baseball pitcher with progressive scapular winging and audible snapping during pitching. Unlike typical posterior-type fractures near the costotransverse joint, imaging revealed a cortical discontinuity precisely at the serratus anterior enthesis, consistent with repetitive traction enthesopathy. High-resolution musculoskeletal ultrasound (MSK-US) identified cortical disruption with periosteal edema, and dynamic ultrasound reproduced the patient’s snapping and pain in real time, establishing a direct clinical–imaging correlation. Conservative three-phase rehabilitation (scapular stabilization, serratus anterior activation, and structured return-to-throwing) led to complete union and pain-free return to sport within 12 weeks. **Discussion:** This case highlights the superior diagnostic efficacy of MSK-US for FRSFs in adolescents. The posterior scanning approach facilitated bilateral comparison and growth plate assessment. Dynamic examination provided a functional correlation beyond static imaging, identifying a novel snapping mechanism. This underscores the value of MSK-US in visualizing not just anatomy but also pathophysiology. **Conclusions:** This is among the youngest documented cases of first rib stress fracture diagnosed with dynamic ultrasound. Its novelty lies in the following: (1) occurrence at the serratus anterior enthesis, (2) reproduction of snapping during provocative maneuvers, and (3) expansion of the etiological spectrum of scapular dyskinesis to include rib pathology. Dynamic ultrasound should be considered a frontline modality for adolescent throwers with unexplained periscapular pain.

## 1. Introduction

Overhead throwing athletes, particularly baseball pitchers, represent a population at an exceptionally high risk for repetitive overuse injuries due to the extreme biomechanical demands inherent to their sport [1,2]. The pitching motion is a complex, high-velocity kinetic chain activity characterized by rapid angular velocities, extreme ranges of motion (e.g., >170° of shoulder external rotation), and the generation of forces that place substantial stress on the shoulder and elbow joints [3,4]. Well-documented sequelae include glenohumeral internal rotation deficit (GIRD), superior labrum anterior–posterior (SLAP) lesions, ulnar collateral ligament (UCL) tears, and posterosuperior internal impingement, which are frequently reported in both adult and adolescent pitchers [5,6]. Consequently, the clinical evaluation of a young throwing athlete often focuses myopically on these areas (Figure 1).

However, the propagation of forces along the kinetic chain also imposes significant stress on the axial skeleton, particularly the ribs. While less common, rib stress fractures represent an important and often missed category of overuse injury. Overall, stress fractures account for up to 20% of all sports-related overuse injuries [7], with rib involvement being relatively uncommon, comprising fewer than 5% of these cases [8]. Among rib injuries, first rib stress fractures hold particular clinical interest due to their unique anatomy and potential for diagnostic confusion. Historically considered rare, these fractures are increasingly recognized in overhead athletes, including baseball players, rowers, swimmers, and weightlifters [9,10,11]. The true incidence is likely underestimated because presenting symptoms—such as vague upper back pain, scapular dyskinesis, or a sensation of snapping—often mimic more common thoracic or scapular disorders like muscular strains, thoracic outlet syndrome, or nerve entrapment neuropathies [12].

Adolescent athletes, in particular, face a “perfect storm” of risk factors: the repetitive high-stress mechanics of throwing combined with a skeleton that is still developing. During growth spurts, long bones lengthen before muscles and tendons have fully adapted, leading to relative inflexibility and increased tensile forces on bone attachments [13,14]. The physeal (growth plate) cartilage is biomechanically weaker than mature cortical bone, making it more susceptible to repetitive microtrauma and avulsion-type injuries [15,16]. Furthermore, neuromuscular control and coordination are still maturing, which can lead to alterations in throwing mechanics that further increase stress on vulnerable structures like the first rib [17]. Despite the implementation of preventive measures like pitch count restrictions, overuse injuries persist in this population, underscoring the limits of current prevention strategies and the need for heightened diagnostic vigilance [18].

Anatomically, the first rib is a unique transitional structure connecting the cervical spine to the thoracic cage. It serves as a critical attachment point for several muscles integral to overhead throwing, including the anterior and middle scalenes, the serratus anterior, and the subclavius, and provides stability via the costoclavicular ligament [19]. Functionally, it acts as a biomechanical crossroad, transmitting forces between the neck and the shoulder girdle. During the late cocking and acceleration phases of pitching, violent contractions of the scalene muscles (to elevate the rib cage) and the serratus anterior (to protract and stabilize the scapula against the thoracic wall) apply intense tensile and torsional stresses across the rib [20,21]. These cyclical loads are concentrated at the rib’s weakest point, the subclavian groove, rendering it highly vulnerable to repetitive microtrauma and eventual fatigue fracture [22].

Traditionally, imaging for suspected rib stress fractures has relied on plain radiographs, computed tomography (CT), and magnetic resonance imaging (MRI). Each modality carries significant limitations. Radiographs are often normal in the early stages due to overlying structures and the subtle nature of initial cortical disruption [12]. While CT provides exquisite cortical detail, it involves substantial ionizing radiation exposure—a single chest CT can deliver an effective dose of 4–8 mSv, equivalent to 100–200 chest X-rays, a major ethical and clinical concern in the pediatric population [23,24]. MRI is highly sensitive for detecting early bone marrow edema and periosteal reaction but is costly, less accessible, often has long wait times, is sensitive to motion artifact, and is unsuitable for patients with certain implants or severe claustrophobia [25]. Most critically, both CT and MRI are static examinations; they provide a snapshot in time but cannot capture the dynamic functional impairments that are often the hallmark of this injury.

Musculoskeletal ultrasound (MSK-US) has emerged as a powerful adjunctive, and in some cases, primary tool in this diagnostic conundrum [26,27]. Its advantages are particularly salient in the pediatric and adolescent population. Beyond being radiation-free, cost-effective, and highly accessible, ultrasound provides a unique interactive dimension. The clinician can directly correlate the imaging findings with the exact point of tenderness through palpation-guided scanning. The ability to perform instantaneous contralateral comparisons is invaluable for identifying subtle asymmetries in cortical continuity or periosteal reaction. Power Doppler mode can further augment the examination by identifying hyperemia associated with active bone stress and periostitis [28]. Most importantly, as this case exemplifies, dynamic ultrasonography allows for the reproduction of functional symptoms under direct visualization, creating an unparalleled clinicoradiological correlation that static imaging cannot offer [29].

While reports of ultrasound-diagnosed first rib stress fractures have increased in collegiate and adult athletes [26], cases describing its application in adolescents remain exceedingly scarce. Moreover, the use of dynamic ultrasound to document functional instability, such as pathologic snapping, has not been thoroughly described in the pediatric population. This case report therefore provides a significant contribution by detailing how dynamic MSK-US was employed not only to identify a cortical fracture but also to dynamically reproduce and elucidate the mechanism of the patient’s symptomatic snapping in a 12-year-old baseball pitcher, advocating for its expanded role as a highly useful adjunct in pediatric sports medicine.

## 2. Case Presentation

### 2.1. Patient History and Presentation

In late April 2019, a 12-year-old Asian male, right-handed, elite-level baseball pitcher presented to our sports medicine clinic for a postural evaluation. The referral was initiated by his concerned coach, who had observed abnormal spinal rotation and asymmetric shoulder positioning during the player’s pitching mechanics, raising initial suspicions of adolescent idiopathic scoliosis.

He reported a several-week history of insidiously progressive right scapular winging, which he first noticed as a “protruding shoulder blade” during his follow-through. More notably, he described a distinct, often painless, audible “snap” or “pop” that consistently occurred during the late cocking phase of his overhead throw. While he denied debilitating pain during activities of daily living, quantifying his discomfort as a mere two out of ten on a visual analog scale, he reported a persistent, dull ache in the right upper thoracic region posteriorly, localized deep to the medial scapular border. Functionally, he expressed growing frustration over a progressive decline in his pitching control and velocity, particularly after pitching three to four innings, accompanied by a subjective sense of weakness and loss of “whipping” power in his throwing arm. His training regimen was intensive, involving pitching practices four times weekly and competitive games on weekends, often exceeding 70 pitches per outing—a volume that approaches and sometimes exceeds recommended guidelines for his age group [30].

### 2.2. Physical Examination Findings

Inspection revealed obvious static and dynamic right scapular dyskinesis. At rest, mild medial border prominence was evident. During dynamic assessment with wall push-ups, pronounced scapular winging with more than 3 cm of medial border displacement was observed, classic for serratus anterior muscle dysfunction. Manual muscle testing revealed significant weakness: grade 3/5 strength (Medical Research Council scale) in scapular protraction (“punch-out” test) and grade 4/5 weakness in the lower trapezius muscle. Palpation of the periscapular region elicited focal, sharp tenderness directly over the posterior aspect of the right first rib, just lateral to its costotransverse junction; this maneuver precisely reproduced the patient’s described upper thoracic discomfort. Cervical spine range of motion was full and pain-free. Neurological examination of the upper extremities was entirely normal, with no sensory deficits, full motor strength in all other myotomes (C5-T1), and symmetric deep tendon reflexes. Specific provocative tests for thoracic outlet syndrome (TOS), including Adson’s test and Wright’s hyperabduction test, were negative, effectively ruling out vascular or neurogenic TOS as the primary cause.

### 2.3. Imaging Findings

Initial diagnostic workup included anteroposterior and oblique cervical spine radiographs. These films effectively ruled out any significant spinal curvature but serendipitously revealed a subtle cortical irregularity and slight undulation along the posterior arch of the right first rib (Figure 2, white arrows). While not diagnostic of a definitive fracture line, this finding was highly suggestive of an underlying stress reaction or early fracture.

Given the radiographic suggestion and high clinical suspicion, a comprehensive musculoskeletal ultrasound examination was performed immediately thereafter at the point of care. Static assessment was conducted with the patient seated upright. Using a high-frequency linear array transducer (L4–20t-RS, 18 MHz; GE Healthcare, Boston, MA, USA) placed in a posterior-to-anterior orientation parallel to the rib axis, a definitive 3.2 mm transverse cortical discontinuity was identified at the site of the serratus anterior muscle attachment on the right first rib. The fracture was accompanied by significant hypoechoic periosteal thickening (elevation) and hyperechoic soft-tissue edema surrounding the breach (Figure 3A). Contralateral comparison scanning of the asymptomatic left first rib confirmed the abnormality by demonstrating an intact, smooth cortical line without periosteal reaction (Figure 3B). To gain a broader contextual view of the scapulothoracic interface, a low-frequency curvilinear probe (C1–5, 3 MHz) was also utilized, which ruled out any associated bursitis or effusion.

### 2.4. Dynamic Ultrasonography: The Pivotal Maneuver

The most diagnostically illuminating part of the examination was the dynamic assessment. The high-frequency linear transducer was again positioned over the posterior first rib/scapular interface. The patient was then instructed to actively and slowly elevate his right arm into abduction and external rotation (ABER), meticulously simulating the late cocking phase of his pitching motion (Figure 4A). Real-time ultrasound imaging captured distinct micromotion at the previously identified fracture site (Figure 4B). Most importantly, this maneuver successfully reproduced the patient’s exact symptomatic, audible snapping sensation. The dynamic clip (Appendix A) clearly demonstrated that the snapping originated from the abrupt catching and release of the fibers of the levator scapulae muscle over the unstable, elevated periosteal fragment at the fracture site on the serratus anterior, providing an undeniable functional correlation between the imaging findings and the clinical complaint.

### 2.5. Therapeutic Intervention and Rehabilitation

Management followed a structured, progressive three-phase rehabilitation protocol grounded in principles of tissue healing and biomechanical restoration:**Phase 1 (Weeks 1–4): Absolute Rest and Protection.** Immediate cessation of all throwing activities was mandated. The arm was placed in a sling for the first 7–10 days to ensure strict rest and minimize pull from the scalene and serratus muscles. Daily physical therapy focused on pain modulation (cryotherapy), and early scapular stabilization exercises began, emphasizing low-load, closed-chain serratus anterior and lower trapezius activation (e.g., scapular protraction in quadruped and wall slides) and postural re-education to address underlying contributory factors.**Phase 2 (Weeks 5–8): Gradual Reloading and Strengthening.** The sling was discontinued. Under guidance, a gradual interval throwing program was initiated, starting at 50% effort from a shortened distance of 15 m, with frequency and intensity increased weekly based on symptom tolerance. Therapeutic kinesiology taping was applied to facilitate scapular posterior tilt and depression, providing proprioceptive feedback and mechanical support during exercises. Strengthening progressed to include open-chain exercises like prone Y/T/W formations, resistance band rows, and serratus anterior punches.**Phase 3 (Weeks 9–12): Sport-Specific Integration and Return to Play.** The rehabilitation focus shifted to restoring high-speed, sport-specific neuromuscular control. This included plyometric drills (medicine ball throws), advanced long-toss programs, and gradual reintroduction to the mound with strict pitch count limits. Criteria for progression included the following: full, pain-free range of motion; no palpable tenderness at the fracture site; resolution of scapular winging during dynamic tasks; and isokinetic strength testing showing less than 10% deficit compared to the non-dominant side.

### 2.6. Follow-Up and Outcome

Serial clinical and sonographic evaluations were performed at 4-week intervals. Ultrasound imaging documented progressive organization of the fracture callus and complete resolution of the periosteal edema and soft-tissue inflammation. At the 12-week mark, follow-up radiographs confirmed solid bridging callus formation across the previous fracture site, indicating radiographic union (Figure 5). Clinically, the patient demonstrated restored, symmetric scapulothoracic mechanics with no residual winging, full pain-free shoulder strength, and no report of snapping. He was subsequently cleared for a full, unrestricted return to competitive pitching. At a 6-month telephone follow-up, he reported successful participation in a full season without any recurrence of symptoms.

## 3. Discussion

This case provides a comprehensive illustration of the diagnostic challenges and innovative solutions associated with first rib stress fractures in adolescent overhead athletes. It underscores the critical importance of moving beyond a narrow focus on the glenohumeral joint and adopting a holistic kinetic chain approach when evaluating these young athletes.

### 3.1. Epidemiology and Pathophysiological Mechanisms

First rib stress fractures remain a rare but important entity in the spectrum of overhead athletic injuries, particularly in skeletally immature adolescents, where they are likely underdiagnosed [8,30]. The biomechanics of throwing place exceptional demands on the first rib. The late cocking and acceleration phases generate a perfect storm of forces: the scalene muscles contract forcefully to elevate and stabilize the first rib, while the serratus anterior contracts powerfully to protract and stabilize the scapula against the thorax [20,21]. This opposing muscular action generates immense tensile and bending stresses across the first rib, concentrated at its biomechanically weakest point, the subclavian groove [22]. In adolescents, this risk is compounded by physiological factors. Open physes are inherently more elastic but less resistant to repetitive shear and tensile forces than mature cortical bone [15]. Furthermore, rapid growth spurts can lead to relative muscle tightness and imbalances, particularly in the posterior shoulder girdle, increasing the strain on the osseoligamentous structures of the thoracic outlet [13,14]. The patient’s recent growth spurt was a significant predisposing factor, creating a mismatch between bone length and muscular flexibility/strength.

### 3.2. Pathophysiology and Mechanistic Insights

The first rib is anatomically predisposed to stress injury. It is short, broad, and tightly curved, rendering it less compliant to repetitive loading than lower ribs [10]. Multiple muscle attachments—scalenes, subclavius, and serratus anterior—subject the rib to competing directional forces. The scalenes exert superior traction, the subclavius transmits compressive load during clavicular depression, and the serratus anterior applies lateral pull during scapular protraction and stabilization [11]. These opposing vectors produce cyclical tensile and torsional stress at focal regions of the rib.

The main sites of occurrence of first rib stress fracture are divided into three types and are divided into groove, interscalene type, and posterior type in order of frequency [30] (Figure 6). Classically, posterior-type FRSF has been reported adjacent to the costotransverse joint, where load is transmitted through the rib–vertebral articulation [12]. Our case diverges from this pattern. The cortical defect was localized precisely at the serratus anterior attachment site, making it best understood as an avulsion-type enthesopathy. Repetitive contraction of the serratus anterior during pitching likely produced traction forces at its enthesis, leading to microtears, periosteal thickening, and eventual cortical disruption. This mechanism represents a tension failure phenomenon rather than vertebral articulation overload. To our knowledge, this enthesis-specific variant of posterior FRSF has rarely been described in the literature.

Furthermore, the first rib’s intimate anatomical relationship with critical neurovascular structures—namely the brachial plexus and subclavian artery/vein—cannot be overlooked. While our patient presented with no signs of thoracic outlet syndrome (TOS), a first rib stress fracture or the accompanying callus formation has the potential to impinge upon these structures, leading to neurogenic or vascular TOS. This underscores the importance of a thorough neurovascular examination in these cases. Dynamic ultrasound with Doppler can be a valuable bedside tool to assess vascular patency if such concerns arise.

The biomechanics of pitching further illuminate this vulnerability. During the late cocking phase, the shoulder is abducted to over 90° and externally rotated beyond 170°, placing the scalene and serratus anterior muscles under maximum tensile load. The acceleration phase then demands rapid, forceful protraction of the scapula, generating immense traction force precisely at the serratus anterior enthesis on the first rib. During acceleration, angular velocities exceed 7000°/s, transmitting kinetic energy across the thorax. Deceleration demands eccentric scapular stabilization, further stressing the serratus anterior enthesis [13]. Over time, these forces accumulate, especially in the setting of adolescent bone immaturity, producing the observed avulsion fracture.

### 3.3. Scapular Dyskinesis and Novel Mechanistic Observations

Scapular dyskinesis is traditionally defined as abnormal motion of the scapula relative to the thorax. Pathophysiology is usually confined to the scapula–thorax interface, primarily involving the subscapularis–serratus anterior plane [14]. Reported causes include fractures, nerve palsy (long thoracic and spinal accessory), muscular imbalance (serratus anterior and trapezius), or fascial disorders such as bursitis [15]. The first rib is not typically included in this framework.

Our case introduces a unique scenario. Dynamic ultrasound revealed snapping between the levator scapulae and serratus anterior at the site of the rib fracture. This finding expands the scope of scapular dyskinesis: rib instability itself disrupted scapulothoracic rhythm, producing a dyskinesis-like presentation. The novelty lies in this functional linkage—scapular snapping originating from rib pathology.

Beyond the bony injury, the observed snapping mechanism likely involves the fascial interfaces between the levator scapulae and serratus anterior. Repetitive microtrauma could lead to fascial thickening or restricted gliding, contributing to the audible snap. Furthermore, the development of myofascial trigger points in the overloaded periscapular muscles could be a concurrent source of pain and a factor in altering normal scapulothoracic rhythm.

A critical unanswered question is whether scapular dyskinesis predisposed the rib to fracture, or whether fracture-induced instability generated secondary dyskinesis. This “cause versus consequence” dilemma is not resolvable from a single case but underscores the bidirectional relationship between rib integrity and scapular mechanics. The observation that rib pathology can mimic dyskinesis suggests that clinicians should reconsider the etiologic spectrum of scapulothoracic abnormalities. Furthermore, the specific location of the fracture at the enthesis provides a clear anatomical target for regenerative interventions like prolotherapy, which aims to strengthen the tendon-bone interface and address this type of traction enthesopathy directly [31].

### 3.4. The Evolving Diagnostic Paradigm: The Central Role of MSK-US

This case powerfully illustrates the evolving role of MSK-US as a powerful first-line diagnostic tool, which can often preclude the need for advanced imaging but remains complementary to MRI or CT in complex or equivocal cases as follows:**Safety:** It completely avoids ionizing radiation, a paramount concern in children and adolescents [23,24].**Accuracy:** It offers high-resolution imaging of cortical bone surfaces, periosteum, and immediate soft tissues, allowing for the detection of subtle fractures and early stress reactions that are often radiographically occult [26,27].**Interactivity:** Its real-time nature facilitates instant side-to-side comparison and palpation-guided assessment, which is invaluable for identifying subtle asymmetries and precisely localizing pathology [28].**Functionality:** Most uniquely, it enables dynamic functional assessment. This is the single greatest advantage over static imaging modalities like CT and MRI. It allows the clinician to not just see the structural defect but to watch it move and to provocatively manipulate the limb to reproduce the patient’s exact symptoms, thereby establishing a direct cause-and-effect relationship [29].

It is crucial to position MSK-US not as a replacement for other modalities, but as an integral part of a multimodal diagnostic workflow that leverages the strengths of each technology to achieve the most accurate diagnosis with the least risk to the patient.

### 3.5. The Diagnostic Power of Dynamic Assessment and a Novel Snapping Mechanism

Our report provides two significant advancements to the literature. First, we detail and advocate for a posterior scanning approach. While most previous reports describe anterior or supraclavicular views, our posterior approach, scanning from the back with the transducer angled cranially, offers distinct advantages: it provides a simultaneous comparative view of both first ribs in a single image frame and offers unparalleled visualization of the posterior rib arch and the costotransverse junction—a common fracture site that is proximate to the growth plates in adolescents.

Second, and more notably, we dynamically identified and documented a previously unreported mechanism for the symptomatic snapping. While scapular snapping is often attributed to scapulothoracic bursitis, osteochondromas, or anomalous muscular slips, our real-time imaging captured the levator scapulae muscle snapping over the unstable, elevated periosteum at the serratus anterior attachment site on the fractured first rib during the ABER maneuver. This finding is crucial. It moves beyond simply diagnosing the fracture to explaining a key functional symptom, thereby validating the patient’s complaint and directly linking the structural injury to the functional impairment. This elevates the diagnostic value of ultrasound from anatomical detection to functional pathophysiological investigation.

### 3.6. A Proposed Diagnostic Algorithm for the Adolescent Thrower

Based on our experience and the literature, we propose the following structured, efficient, and safe diagnostic algorithm for the adolescent overhead athlete with unexplained periscapular or upper thoracic pain (Figure 7). Figure 7 provides a visual summary of this proposed diagnostic pathway. This algorithm prioritizes a radiation-free, point-of-care pathway that integrates clinical examination with progressive tiers of ultrasonographic assessment.

**Step 1: Comprehensive History and Physical Exam:** A meticulous history and physical exam are the cornerstones of diagnosis. Key historical red flags include mechanical symptoms like reproducible snapping or popping, pain localized to the superior medial scapular border or supraclavicular fossa, and symptoms exacerbated by the late cocking or acceleration phases of throwing. The physical exam must include:Inspection: For static and dynamic scapular dyskinesis.Palpation: Focal tenderness over the posterior first rib is a paramount finding.Neuromuscular Exam: Manual muscle testing of the serratus anterior (punch-out test) and lower trapezius.Provocative Tests: Specific tests for thoracic outlet syndrome (Adson’s, Wright’s) and cervical radiculopathy should be performed to rule out neurogenic and vascular causes.**Step 2: Point-of-Care Static Ultrasound:** This should be the first-line imaging modality. Using a high-frequency linear transducer, a systematic examination is performed:Primary Target: Scan the posterior arch of the symptomatic first rib in a craniocaudal and axial plane to assess for cortical discontinuity, periosteal thickening (elevation), or hypoechoic edema.Contralateral Comparison: An immediate scan of the asymptomatic side is mandatory to establish the patient’s normal anatomy and identify subtle asymmetries.Soft-Tissue and Enthesis Evaluation: Examine the serratus anterior enthesis for signs of enthesopathy (e.g., calcifications, cortical irregularity, and thickening), which may serve as an early biomarker for stress. Use a low-frequency curvilinear probe to assess the scapulothoracic space for bursitis or effusion.Doppler Interrogation: Power Doppler mode should be employed to detect hyperemia associated with active bone stress reaction or periostitis, adding a functional component to the static exam.**Step 3: Dynamic Ultrasonographic Provocation:** This is the critical step for functional diagnosis. If clinical suspicion remains high or a functional symptom is reported, the static exam must be followed by dynamic assessment.Maneuver: Under direct ultrasound guidance, the patient actively and slowly performs the symptomatic movement (e.g., active arm abduction and external rotation (ABER) to simulate the late cocking phase).Objective: The goal is to dynamically visualize micromotion at the fracture site and, most importantly, to reproduce the patient’s exact snapping sensation. This provides irrefutable clinicoradiological correlation and elucidates the pathomechanism (e.g., muscle snapping over a periosteal fragment).**Step 4: Advanced Cross-Sectional Imaging (MRI):** MRI is reserved for specific scenarios where ultrasound is as follows:Technically difficult or equivocal.Negative despite a high clinical suspicion.There is a need to rule out alternative pathologies with similar presentation (e.g., osteoid osteoma, osteomyelitis, and bone cyst).MRI excels at confirming bone marrow edema, a hallmark of stress reaction, and provides a comprehensive overview of all surrounding soft tissues.

### 3.7. Comprehensive Management and Future Directions

Conservative management, as outlined in our case, remains the cornerstone of treatment for uncomplicated FRSF, with excellent outcomes typically achieved within 3 months [31,32]. The principles involve activity modification to unload the fracture, progressive rehabilitation to address underlying biomechanical deficits (especially serratus anterior and lower trapezius strength and endurance), and a graded, criteria-based return to sport.

For refractory cases or those progressing to non-union, minimally invasive interventional techniques such as prolotherapy or extracorporeal shockwave therapy (ESWT) represent potential areas for future research [31,33,34]. For cases with significant associated myofascial pain, techniques such as ultrasound-guided dry needling could be considered as a potential adjunct to rehabilitation to alleviate pain and improve muscle function, though further research is needed in this specific population. The principles of conservative management involve activity modification to unload the fracture, progressive rehabilitation to address underlying biomechanical deficits (especially serratus anterior and lower trapezius strength and endurance), and a graded, criteria-based return to sport. The role of other modalities requires further controlled investigation in athletes with persistent symptoms before any clinical recommendations can be made.

### 3.8. Limitations

This study has limitations inherent to a single case report. Its findings, while compelling, need validation in larger prospective studies. Musculoskeletal ultrasound is operator-dependent, and the technical skill required to visualize the first rib, especially via the posterior approach, requires training and practice. The dynamic examination protocol, while highly informative, is not yet standardized. Factors such as the degree of arm abduction and external rotation and the precise transducer orientation can vary between operators. Future studies should aim to develop and validate a standardized dynamic protocol for assessing scapulothoracic and first rib dynamics to improve inter-observer reliability. Finally, our follow-up period, while sufficient to document healing, does not provide long-term data on recurrence rates or the risk of subsequent injury.

## 4. Conclusions

This case report profoundly reinforces the diagnostic utility of musculoskeletal ultrasound for first rib stress fractures in adolescent athletes and breaks new ground by showcasing the unique functional insights afforded by dynamic assessment. We advocate for a high index of suspicion for this injury in young overhead throwers presenting with periscapular pain, winging, or mechanical snapping. When clinical suspicion arises, point-of-care ultrasound should be employed early in the diagnostic pathway. The described posterior scanning technique enhances the ability to assess the fracture and compare bilaterally. Most importantly, incorporating dynamic maneuvers to replicate the athletic motion and reproduce symptoms transforms the examination, providing irrefutable clinicoradiological correlation and offering insights into the functional pathophysiology of the injury, as demonstrated by the novel levator-scapulae/serratus-anterior snapping mechanism we identified.

We posit that dynamic musculoskeletal ultrasonography is a powerful, non-invasive, and highly informative tool that deserves broader integration into the standard diagnostic algorithm for pediatric and adolescent athletes with overhead throwing injuries. It moves the diagnostic process from inferring a correlation based on static images to directly observing the cause of dysfunction in real-time. This not only ensures accurate diagnosis but also enhances patient understanding and adherence to the often-frustrating process of rest and rehabilitation. As technology and expertise proliferate, dynamic MSK-US should become a cornerstone in the evaluation of the young throwing athlete.

## Figures and Tables

**Figure 1 diagnostics-15-02437-f001:**
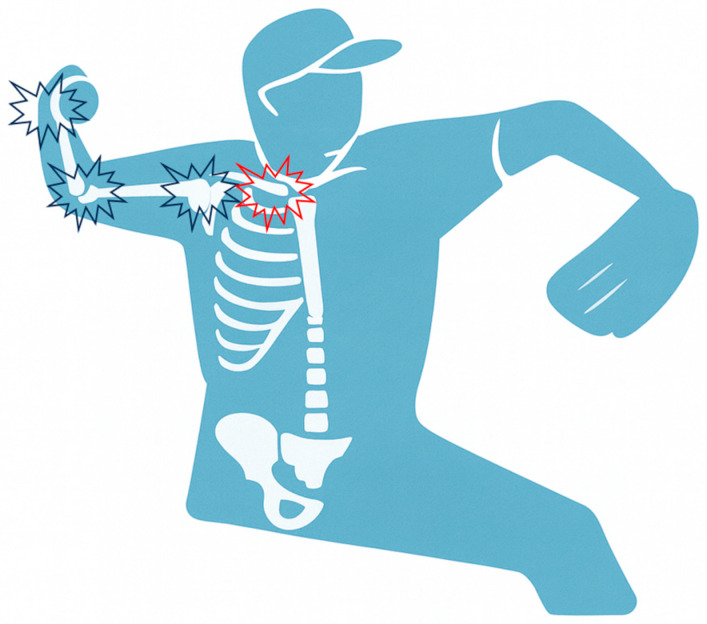
Schematic diagram of pitching form and common injury locations in upper extremities. Black stars indicate widely known problems, while red stars indicate damage that is often overlooked.

**Figure 2 diagnostics-15-02437-f002:**
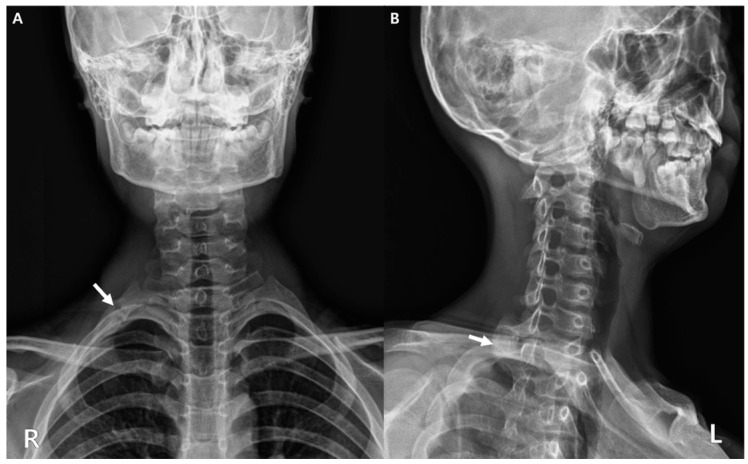
Anteroposterior (**A**) and oblique (**B**) cervical spine radiographs showing a cortical irregularity of the right first rib (white arrow). R, right; L, left.

**Figure 3 diagnostics-15-02437-f003:**
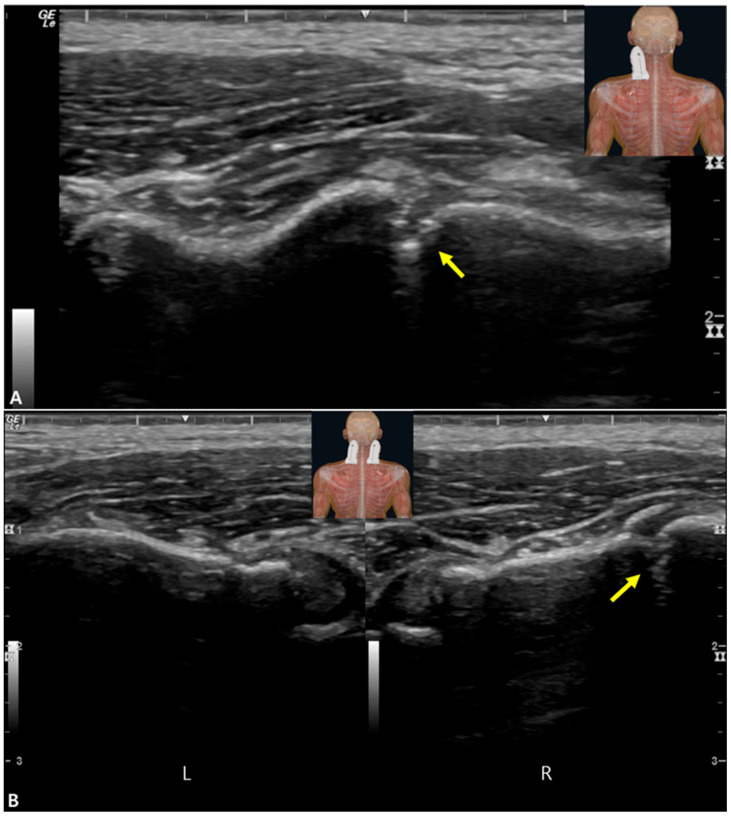
(**A**) Musculoskeletal ultrasound demonstrating a focal cortical discontinuity of the right first rib with adjacent periosteal thickening and soft-tissue edema (yellow arrow). (**B**) Contralateral side for comparison without cortical defect. R, right; L, left.

**Figure 4 diagnostics-15-02437-f004:**
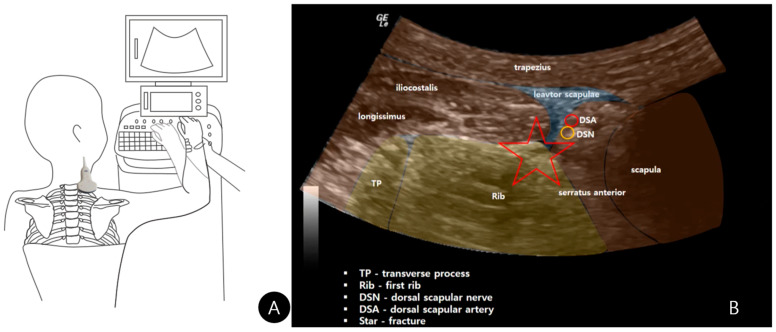
Dynamic ultrasound examination. (**A**) Patient and probe positions during the provocative maneuver. (**B**) Still image highlighting the fracture site (star) at the time of symptom reproduction.

**Figure 5 diagnostics-15-02437-f005:**
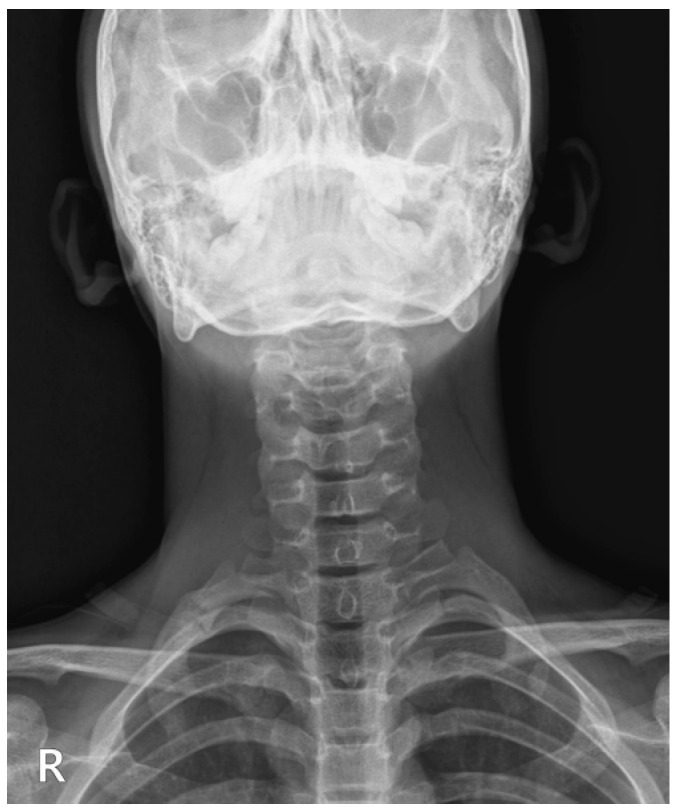
Follow-up anteroposterior cervical spine radiograph demonstrating callus formation and healing of the right first rib compared with the initial image. R, right.

**Figure 6 diagnostics-15-02437-f006:**
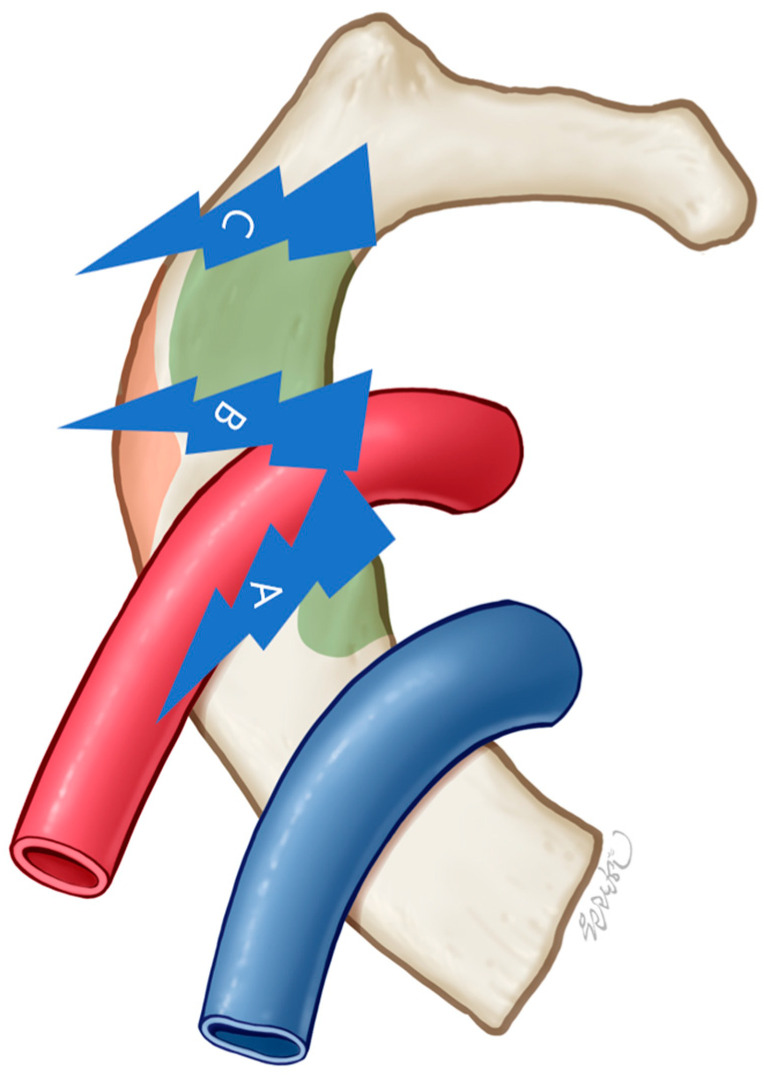
Schematic diagram of the anatomy of the first rib and common fracture sites. (A) groove type, (B) interscalene type, (C) posterior type.

**Figure 7 diagnostics-15-02437-f007:**
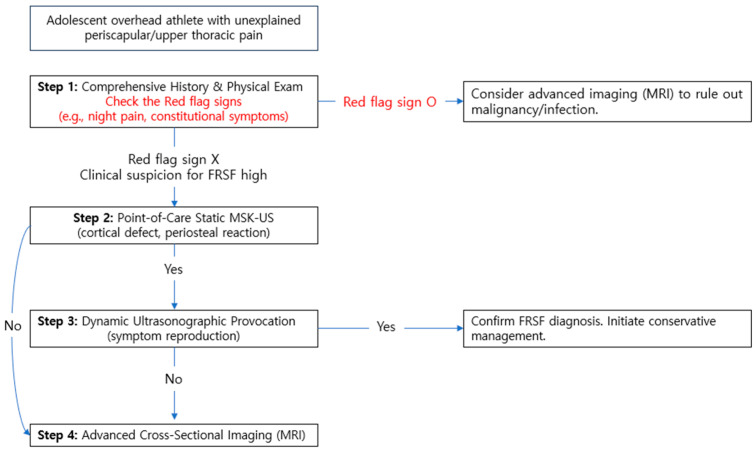
Proposed clinical and ultrasonographic diagnostic algorithm for adolescent overhead athletes with suspected first rib pathology. Step 1 excludes red flag signs (night pain, constitutional symptoms).

## Data Availability

The original contributions presented in this study are included in the article. Further inquiries can be directed to the corresponding authors.

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
