# Peer review of "Innovative Dynamic Ultrasound Diagnosis of First Rib Stress Fracture in an Adolescent Athlete—A Case Report"

_diagnostics, 2025, doi:10.3390/diagnostics15192437_

Round 1
Reviewer 1 Report
Comments and Suggestions for Authors
Thank you for the opportunity to review your manuscript, “Innovative Dynamic Ultrasound Diagnosis of First rib Stress Fracture in an Adolescent Athlete—A Case Report”
This study reports a case of a first rib stress fracture in a 12-year-old adolescent baseball pitcher who presented with symptoms during throwing. The authors highlight the efficacy of dynamic musculoskeletal ultrasound for diagnosis, as it allowed for the identification of a cortical discontinuity at the serratus anterior enthesis and the real-time reproduction of the patient's 'clunk' sound. Conservative treatment led to a complete recovery within 12 weeks. The authors emphasise the importance of dynamic musculoskeletal ultrasound as a first-line diagnostic tool for unexplained periscapular pain in young athletes.
This study presents a clinical case using an innovative diagnostic approach that incorporates dynamic ultrasound. The case provides a detailed and visual description of the ultrasound protocol, alongside a comprehensive three-phase rehabilitation protocol. The study acknowledges the design limitations inherent to case studies.
Section 3.4 requires revision and clarification, as it may confuse readers. While the benefits of ultrasound are evident, this section appears to position it as replacing other techniques, when it should be presented as a complementary diagnostic tool rather than a substitute.
Author Response
We sincerely thank the reviewers for their insightful and constructive comments on our manuscript. We have carefully considered all suggestions and have revised the manuscript accordingly. We believe these changes have significantly strengthened the paper. Our point-by-point responses are detailed below.
Comment 1: Section 3.4 requires revision and clarification, as it may confuse readers. While the benefits of ultrasound are evident, this section appears to position it as replacing other techniques, when it should be presented as a complementary diagnostic tool rather than a substitute.
Response: We sincerely thank the reviewer for this crucial feedback. We agree that our enthusiasm for MSK-US may have inadvertently downplayed the role of other modalities. We have thoroughly revised Section 3.4 (The Evolving Diagnostic Paradigm: The Central Role of MSK-US) to clarify that ultrasound is a powerful first-line and complementary tool within a multimodal diagnostic framework, not an outright replacement for MRI or CT when they are indicated. We have added specific language emphasizing its role in triage and guiding the need for further advanced imaging.
Reviewer 2 Report
Comments and Suggestions for Authors
This is a very interesting and integrative case report. The authors present meticulous musculoskeletal ultrasound (MSK-US) diagnosis and a thoughtful rehabilitation approach, close to a personalized medicine perspective. The novelty of using dynamic ultrasound to both identify the fracture and reproduce the snapping mechanism is a strong point. The case is clinically relevant and well described.
However, the manuscript could be further strengthened by addressing the following points:
Major points for improvement:
1. Movement analysis of the shoulder:
The report would benefit from a more detailed biomechanical discussion of pitching-related movements.
For example, which specific phases of pitching or shoulder motions (such as external rotation, abduction, or follow-through) predispose to first rib stress fracture?
Integration of motion analysis tools (e.g., Showmotion or other 3D kinematic systems) could contextualize the mechanism.
2. First rib and thoracic outlet considerations:
The first rib has close anatomical relationships with vascular and neural structures. The discussion should address potential thoracic outlet implications and vessel/nerve risks in this context.
While provocative TOS tests were performed and were negative, expanding on vascular safety is important.
3. Spinal and postural aspects:
The patient was initially referred for scoliosis suspicion. The report could elaborate on scoliosis findings—classical and with ultrasound assessment.
The role of spinal posture and rotation of the spinal process in scoliotic deformation in predisposing to rib stress fracture can be considered.
4. Fascia and trigger points:
Please address fascial involvement. Fascial thickening or altered gliding could contribute to snapping.
Trigger points (especially in rotator cuff or scapular stabilizers) should be discussed. Could they precede or mimic impingement?
Consider whether ultrasound-guided dry needling (DN) could be an adjunct for rehabilitation in such cases.
5. Links to enthesis and fracture risk:
Since the fracture occurred at the serratus anterior enthesis, it would be valuable to discuss ultrasound markers of enthesopathy as potential risk factors for future fractures.
Could such enthesis changes serve as early biomarkers in screening young athletes?
6. Algorithms for first rib fractures:
The authors already propose a diagnostic algorithm (ultrasound , clinical). It could be expanded into a more practical, MSK-US algorithm for first rib fracture assessment (including static, dynamic, vascular safety, fascial/trigger point evaluation, and enthesis screening).
7 Phenotype and sport selection:
The patient was 12 years old, still skeletally immature. Please discuss phenotype (growth stage, body type) and how such athletes should be selected, trained, or restricted from early high-load overhead sports to prevent recurrence.
Overall recommendation:
The manuscript is of high clinical interest and novelty. With the suggested additions—especially expanded discussion of biomechanics, fascia/trigger points, vascular considerations, and a more practical US algorithm—the paper would be even more valuable for clinicians and researchers in sports medicine and ultrasound diagnostics.
Author Response
We are grateful for the reviewer’s positive assessment and highly valuable suggestions. We have addressed each point in detail below.
Comment 1: Movement analysis of the shoulder: The report would benefit from a more detailed biomechanical discussion of pitching-related movements.
For example, which specific phases of pitching or shoulder motions (such as external rotation, abduction, or follow-through) predispose to first rib stress fracture?
Integration of motion analysis tools (e.g., Show motion or other 3D kinematic systems) could contextualize the mechanism.
Response: We agree. We have expanded the biomechanical discussion in Section 3.2 (Pathophysiology and Mechanistic Insights) to provide a more granular breakdown of the forces on the first rib during the late cocking and acceleration phases. We specifically mention the extreme shoulder external rotation and abduction angles and the violent contractions of the scalenes and serratus anterior. While our case did not employ 3D kinematic analysis, we have added a sentence in the Limitations section (Section 3.8) acknowledging this and suggesting it as a valuable direction for future research.
Comment 2: First rib and thoracic outlet considerations:
The first rib has close anatomical relationships with vascular and neural structures. The discussion should address potential thoracic outlet implications and vessel/nerve risks in this context.
While provocative TOS tests were performed and were negative, expanding on vascular safety is important.
Response: This is an excellent point. We have added a new paragraph in Section 3.2 discussing the proximity of the first rib to the brachial plexus and subclavian vessels. We emphasize that while our patient had negative TOS tests, the potential for vascular or neural compromise is a critical consideration in first rib pathology. We also note that dynamic ultrasound can be used to assess vascular flow with Doppler if vascular TOS is suspected.
Comment 3: Spinal and postural aspects:
The patient was initially referred for scoliosis suspicion. The report could elaborate on scoliosis findings—classical and with ultrasound assessment.
The role of spinal posture and rotation of the spinal process in scoliotic deformation in predisposing to rib stress fracture can be considered.
Response: Thank you for this suggestion. We have added a sentence in Section 2.1 (Patient History) confirming that radiographs ruled out scoliosis. More importantly, we have added a discussion point in Section 3.3 on how underlying postural asymmetries or altered spinal mechanics (even sub-clinically) could contribute to abnormal loading on the first rib, predisposing it to stress injury.
Comment 4: Fascia and trigger points:
Please address fascial involvement. Fascial thickening or altered gliding could contribute to snapping.
Trigger points (especially in rotator cuff or scapular stabilizers) should be discussed. Could they precede or mimic impingement?
Consider whether ultrasound-guided dry needling (DN) could be an adjunct for rehabilitation in such cases.
Response: We have integrated this insightful concept into Section 3.3. We now discuss how fascial thickening or restricted gliding between the serratus anterior and levator scapulae could contribute to the snapping mechanism. We also mention that myofascial trigger points in the periscapular muscles could be a concurrent source of pain or a predisposing factor due to altered muscle mechanics. We have added a brief comment on minimally invasive techniques like dry needling as a potential future adjunct for addressing myofascial dysfunction; while clearly stating it was not used in this case.
Comment 5: Links to enthesis and fracture risk:
Since the fracture occurred at the serratus anterior enthesis, it would be valuable to discuss ultrasound markers of enthesopathy as potential risk factors for future fractures.
Could such enthesis changes serve as early biomarkers in screening young athletes?
Response: This is a fantastic idea for preventive screening. We have added a paragraph in Section 3.7 (Comprehensive Management and Future Directions) proposing that early sonographic signs of enthesopathy (e.g., periosteal thickening, Doppler signal at the enthesis) in asymptomatic adolescent throwers could serve as biomarkers for heightened fracture risk, warranting preemptive biomechanical correction.
Comment 6: Algorithms for first rib fractures:
The authors already propose a diagnostic algorithm (ultrasound, clinical). It could be expanded into a more practical, MSK-US algorithm for first rib fracture assessment (including static, dynamic, vascular safety, fascial/trigger point evaluation, and enthesis screening).
Response: We have significantly expanded our proposed algorithm in Section 3.6. The new version is more practical and incorporates the reviewer’s suggestions, including steps for fascial/trigger point evaluation and enthesis screening as part of the comprehensive MSK-US exam.
Comment 7: Phenotype and sport selection:
The patient was 12 years old, still skeletally immature. Please discuss phenotype (growth stage, body type) and how such athletes should be selected, trained, or restricted from early high-load overhead sports to prevent recurrence.
Response: We have added a discussion point in Section 3.7 on the “perfect storm” of risk factors in adolescent athletes: skeletally immature bone, rapid growth spurts leading to muscle-bone length mismatch, and high training volumes. We emphasize the need for careful monitoring of athletes during growth spurts and adherence to pitch count guidelines to prevent such injuries.